# A Psychophysical Window onto the Subjective Experience of Compulsion

**DOI:** 10.3390/brainsci11020182

**Published:** 2021-02-02

**Authors:** Stefan Schmidt, Gerd Wagner, Martin Walter, Max-Philipp Stenner

**Affiliations:** 1Department of Psychiatry and Psychotherapy, Jena University Hospital, 07743 Jena, Germany; stefan.schmidt@uni-jena.de (S.S.); wagner.gerd@uni-jena.de (G.W.); Martin.Walter@med.uni-jena.de (M.W.); 2Department of Behavioral Neurology, Leibniz Institute for Neurobiology, 39118 Magdeburg, Germany; 3Department of Neurology, Otto-von-Guericke University, 39120 Magdeburg, Germany

**Keywords:** obsessive–compulsive disorder, sense of agency, intentional binding, cue integration

## Abstract

In this perspective, we follow the idea that an integration of cognitive models with sensorimotor theories of compulsion is required to understand the subjective experience of compulsive action. We argue that cognitive biases in obsessive–compulsive disorder may obscure an altered momentary, pre-reflective experience of sensorimotor control, whose detection thus requires an implicit experimental operationalization. We propose that a classic psychophysical test exists that provides this implicit operationalization, i.e., the intentional binding paradigm. We show how intentional binding can pit two ideas against each other that are fundamental to current sensorimotor theories of compulsion, i.e., the idea of excessive conscious monitoring of action, and the idea that patients with obsessive–compulsive disorder compensate for diminished conscious access to “internal states”, including states of the body, by relying on more readily observable proxies. Following these ideas, we develop concrete, testable hypotheses on how intentional binding changes under the assumption of different sensorimotor theories of compulsion. Furthermore, we demonstrate how intentional binding provides a touchstone for predictive coding accounts of obsessive–compulsive disorder. A thorough empirical test of the hypotheses developed in this perspective could help explain the puzzling, disabling phenomenon of compulsion, with implications for the normal subjective experience of human action.

## 1. Introduction

Obsessive–compulsive disorder (OCD) is characterized by obsessions, compulsions, or both. Compulsions include repetitive overt behavior, such as checking, or repetitive mental acts, such as counting, which a person performs in response to an obsession, or following a rigid rule, aiming to reduce distress, or to prevent a dreaded event [1]. Importantly, a defining feature of compulsions is a subjective experience of being driven to these behaviors or mental acts, i.e., a subjective loss of control [2]. Compulsive checking, for instance, is characterized by an overwhelming drive towards seemingly re-assuring, yet unwanted action, a drive that leaves little or no choice. At the same time, patients with OCD often assume, at least implicitly, having more causal influence on the world [3] than they actually do, e.g., an influence on future catastrophic events [4,5], such as magical thinking. Why do patients with OCD experience a loss of control over their own choice, and, at the same time, assume a particularly strong causal influence on the world?

In this perspective, we follow the idea that an answer to these questions, and one way to gain insight into the subjective experience of compulsive action, lies in an integration of classic cognitive models of OCD with contemporary sensorimotor theories of compulsion [6], reviewed in Section 2. We argue that cognitive biases in OCD may obscure an altered momentary, pre-reflective experience of sensorimotor control, at least when experimental observation relies purely on patients’ explicit reports of control, e.g., on explicit judgements of agency. To gain insight into this pre-reflective experience, an implicit experimental operationalization of the subjective experience of action and agency is needed instead, one that is more robust to any obscuring effects of cognitive biases than explicit reports to overt inquiry.

We argue that a classic psychophysical test exists that provides this implicit operationalization. This test is the intentional binding paradigm introduced by Haggard et al. [7]. Moreover, we show that intentional binding can pit two ideas that are fundamental to current sensorimotor theories of compulsion against each other. These are the idea of excessive conscious monitoring of action in OCD [8,9,10,11] and the idea that patients with OCD compensate for diminished conscious access to “internal states”, including states of the body, by relying on more readily observable proxies [8,12]. These ideas make distinct predictions regarding the cause of an altered experience of action and agency in OCD, and ultimately differ in their implications for therapeutic approaches. We show that intentional binding provides an opportunity to test these predictions, and a useful window onto an altered subjective experience of action and agency in OCD. While some aspects of intentional binding have been tested in a non-clinical population of healthy volunteers, divided into subgroups with relatively high vs. low obsessive–compulsive tendencies [13], the full potential of intentional binding to discriminate between ideas that are fundamental to current theories of OCD has remained unexploited.

We first provide a brief overview on theoretical considerations and empirical evidence in support of the idea that compulsive action is driven, or at least maintained, by an altered subjective experience of sensorimotor function, over and above an undisputed role of cognitive biases. Turning to two ideas that are fundamental to current sensorimotor theories of compulsion, we develop concrete, testable hypotheses regarding altered intentional binding under the assumption of different sensorimotor accounts of compulsion. We discuss to what extent previous experimental work has, and to what extent it has not yet, addressed these hypotheses. Finally, we discuss limitations of intentional binding as a measure of the subjective experience of action and agency in OCD.

## 2. Altered Experience of Sensorimotor Function in OCD

For several decades, cognitive and metacognitive biases have been a major focus in OCD research. Research into dysfunctional beliefs has provided insight into potential mechanisms of symptom generation and/or maintenance, as well as clinical heterogeneity in OCD [5,14], with implications for treatment [15]. In subgroups of patients with OCD [16], cognitive biases may include an overestimation of threat or vulnerability [17], inflated assumption of responsibility [4], overestimation of the significance of thoughts, including beliefs that a mere thought changes the likelihood of events in the world, or is morally equivalent to carrying out a corresponding action [18], intolerance of uncertainty, and an exaggerated strive for perfection [5].

However, several authors have argued that cognitive biases alone may not sufficiently account for the full spectrum of possible symptoms, and, in particular, may not explain specific phenomenal features of compulsive action. Levy [9], for example, motivates his alternative focus on predictive coding in OCD by previous descriptions of patient subgroups who do not differ in (reported) cognitive biases from healthy individuals [19,20]. Along similar lines, Szalai [6] argues that dysfunctional beliefs do not explain two common features of compulsive action, specifically, the repetitive nature of compulsive behavior, and a feeling of incompleteness of action [21], known to be statistically (at least partly) independent from certain cognitive beliefs [22]. Incompleteness refers to a feeling that an action or intention has not (yet) been properly achieved, a feeling that many patients with OCD have experienced [23].

To accommodate these behavioral and phenomenal features of compulsive action, several authors have argued in favor of altered sensorimotor function, or altered subjective experience of sensorimotor function, in addition to, and interacting with, cognitive biases. These accounts converge on two key ideas: the idea that patients attempt to monitor their actions differently from healthy individuals [8,9,10,24], and the idea that patients have diminished conscious access to “internal states”, for which they compensate by relying on observable proxies (see below for a definition of these states in the original model [8,12]). We review these accounts, together with their implications for treatment of OCD, below and in Table 1.

### 2.1. Abnormal Action Monitoring

Several authors have proposed that patients with OCD differ from healthy individuals in their (conscious) monitoring of, and/or attentional focus on, their own actions [8,9,10,24]. De Haan et al. [10], for example, propose that patients with OCD exert more conscious, deliberate, and reflective control over their own actions, “trying to perform all actions with maximal attention” [25], compared to healthy individuals, who, in turn, are more often pre-reflectively “immersed in […] action”. Following the idea of a “hyper-reflexivity trap” [31], de Haan et al. [10] argue that excessive conscious control, e.g., with the goal to reduce uncertainty regarding action completion [25], augments, rather than reduces, insecurity, and thus further amplifies conscious control, driving compulsive repetition in search of re-assurance. Accordingly, improvement of OCD symptoms, e.g., under deep brain stimulation [25], should be accompanied by a reduction in conscious control over (habitual) action [10].

In support of the idea of altered monitoring of action in OCD, Belayachi and Van der Linden [24,26] have provided evidence that compulsive checking may be accompanied by an altered focus in perceiving one’s own actions. In their study, checking symptoms in a non-clinical population were associated with a tendency to describe actions in terms of their motoric and mechanistic details, rather than their overarching goals [26]. In reference to Vallacher’s and Wegner’s Action Identification Theory [32], the authors suggested that this “low-level identification” of action may promote the detection of “inconsistency” and error, and possibly impair “reality monitoring”, i.e., discrimination of the intended or imagined from the actually performed action [33]. Indeed, patients with OCD show enhanced electrophysiological responses to performance errors (e.g., Gehring et al. [34]). Enhanced error-related negativity in electroencephalography (EEG) has been proposed as a robust endophenotype of OCD [35], indexing increased sensitivity to errors that may trigger explicit control processes and lead to an error- or harm-avoidant response style associated with tension or distress/anxiety.

Finally, and in line with the idea of altered monitoring of action in OCD, Levy [9] has proposed a predictive coding account of OCD, emphasizing a role of “attention to normally unattended sensory and motor representations”. Levy [9] argues that such undue attention in OCD alters the assignment of precision, i.e., certainty, to prior beliefs and sensory evidence (see also Kiverstein et al. [36]), an idea to which we will return in the next two sections.

Similar to the above reviewed accounts by de Haan [10], Belayachi and Van der Linden [11], and Levy [9], Liberman and Dar [8] also assume “a cycle of ever-increasing […] monitoring” in OCD, specifically monitoring of a patient’s progress towards their goals. However, in addition to enhanced monitoring attempts, Liberman and Dar [8] argue that patients with OCD also have monitoring difficulties.

### 2.2. Diminished Conscious Access to “Internal States”: The SPIS Model

Following Shapiro’s [27] idea of a “loss of conviction” in OCD, Liberman and Dar [8] propose that patients with OCD have diminished (conscious) access to “internal states”, which the authors define to include feelings, wishes, and preferences, but also momentary states of the body, in particular as sensed via proprioception [12,28,30]. Importantly, Liberman and Dar [8] assume that patients compensate for this diminished access by relying strongly on “proxies” that are more readily observable, such as behavioral rituals or sensory input from the environment. This is called the “seeking proxies for internal states” (SPIS) model of OCD.

Interestingly, while Liberman and Dar’s definitions of “internal states” and of potential “proxies” for these states are both broad and include such diverse categories as wishes, proprioception, and rules, empirical evidence for the SPIS model has come predominantly from studies which examine conscious access to states of sensorimotor systems, and which focus on external stimulation as a putative proxy. For example, in a series of studies, Lazarov et al. [12,29,30] demonstrated that high obsessive–compulsive tendencies in a non-clinical cohort, as well as in subjects diagnosed with obsessive–compulsive disorder, correlated with an over-reliance on visual feedback when participants were asked to produce certain levels of muscle tension, as well as when explicitly estimating and reporting their own muscle tension. Further support for the idea of an imbalance in explicit access to proprioception vs. vision comes from a study by Ezrati et al. [28], who provide evidence that high obsessive–compulsive tendencies may correlate with an over-reliance on visual over proprioceptive feedback when that feedback is used to correct arm movements in a visuomotor rotation paradigm.

While all of the above accounts converge on the idea that compulsive action is characterized by altered explicit monitoring, the SPIS model has potential implications for therapy that are distinct from the other accounts. Following the SPIS model, Lazarov et al. [12,30] argue that training to more accurately monitor “internal states”, e.g., training to monitor muscle tension via bio-feedback, may benefit patients with OCD. In contrast, de Haan et al. [10] follow a very different rationale when they argue that too much conscious monitoring, rather than too little (or misguided) conscious access, is driving compulsive action. According to de Haan et al. [10] successful treatment requires attenuating, rather than training, conscious monitoring of action.

### 2.3. Altered Function of Predictive Models

Feelings of incompleteness, which can accompany compulsive action (e.g., Taylor et al. [22]), may be explained by a very specific sensorimotor monitoring dysfunction, related to predictive internal models. The amplitude of electroencephalographic (EEG) potentials in response to a sensory, e.g., visual, stimulus is smaller when that stimulus is a consequence of one’s own action, e.g., a key press, compared to a condition when the same stimulus is passively viewed [37]. This attenuation effect when a stimulus is triggered by one’s own action has been explained by assuming that an internal model of the contingency between action and stimulus predicts, and thereby “cancels”, actual sensory feedback [38] (but see Brown et al. for an alternative (though predictive) mechanism of sensory attenuation, based on a modulation of precision [39]). Gentsch et al. [40] found that the attenuation of evoked potential amplitude when a visual stimulus was triggered by the participant’s key press, compared to a condition when it was passively viewed, was less pronounced in patients with OCD. The authors interpreted this difference between patients and controls as a sign that the formation of internal models, and thus the above “cancellation”, are altered in OCD. This idea has attracted considerable attention in Szalai’s attempts to integrate sensorimotor function and cognitive biases to describe the sense of agency in OCD [6]. We will return to the idea of altered function of predictive models in OCD in the next two sections.

### 2.4. Phenomenology and Monitoring

A key challenge is to understand how exactly undue conscious monitoring of action [10,24], over-reliance on proxies [8,29], and/or altered function of internal predictive models [6,40], changes the subjective experience of action and agency in OCD. More specifically, the challenge is to obtain a useful measure of the subjective experience of action and agency in a disorder often characterized by strong cognitive biases. Explicit judgements in response to overt inquiry, including judgements of agency, are likely influenced by such cognitive biases [41]. Gentsch et al. [40], for example, interpreted a trend they observed towards enhanced explicit agency judgements in patients with OCD as a result of a classic cognitive bias to assume inflated responsibility (see also Belayachi and Van der Linden [11]). Furthermore, post-hoc introspective reports, including, in theory, judgements of agency, can be subject to retrospective confabulation [42]. Hence, in a disorder in which beliefs are considered to have a particularly strong influence on subjective experience and behavior [14], explicit reports in response to overt inquiry may not reveal more than a confirmation that these cognitive biases exist, rather than describe a momentary, pre-reflective subjective experience.

Consequently, interpretation of several previous studies into the subjective experience of action and agency in OCD, which were based on explicit reports to overt inquiry, is complicated by a potential confound of a (pre-reflective) experience of sensorimotor function with cognitive biases [24,26,43]. Other studies that (partly) avoid explicit judgements, and instead focus on purely physiological measures, such as the amplitude of evoked responses, on the other hand, require strong, somewhat speculative assumptions of a relation between evoked responses in EEG and the subjective experience of action and agency [40].

Thus, the current OCD literature calls for an experimental operationalization of the subjective experience of action and agency that is implicit, and therefore less prone to the obscuring effect of cognitive biases. Ideally, this operationalization would enable the testing—and pitting against each other— of ideas regarding excessive conscious monitoring, diminished conscious access to “internal states”, compensatory over-reliance on observable proxies, and altered internal model function in OCD, as reviewed above.

## 3. Intentional Binding: A Window onto the Subjective Experience of Action

We propose that a psychophysical phenomenon called intentional binding can provide a useful experimental operationalization in this regard. Intentional binding refers to a contraction of the perceived time interval between one’s own motor output and consequent sensory input [7]. For example, the time of a tone is perceived earlier when that tone is caused by a voluntary action, as compared to a condition in which the same tone is presented without voluntary action, i.e., perceived passively, or a condition in which an involuntary movement of the same muscles triggers the tone [7]. The perceived time of a voluntary action, in turn, is bound forward, towards the time of a delayed sensory consequence of that action, e.g., towards a tone triggered by that action after a short delay. Together, these separable phenomena—called “tone binding” and “action binding” in the literature—lead to a contraction of the perceived delay between a voluntary action and its sensory consequence.

Because intentional binding (sometimes called temporal binding) occurs under voluntary, operant conditions, it has frequently been regarded as a phenomenon associated with a sense of agency (see Moore and Obhi [44] for a review). Intentional binding requires both causality and intentionality [45], it depends on the contingency between action and consequence [46], and on prior beliefs of agency [47], and it is attenuated when a person is not free in their choice [48].

Intentional binding can be assessed using a standard psychophysical approach based on the “Libet experiment” [49]. Specifically, participants watch a clock hand rotating at around 0.4 Hz (Figure 1A). In the classic version of the paradigm, there are four conditions. In two conditions, called operant conditions, participants press a button at a time of their own choice and then hear a tone 250 ms later. In separate blocks, they then report the clock hand position they perceived either at the time when they pressed the button, or at the time when they heard the tone. These time judgements are compared to time judgements when the action and the tone are present in isolation, i.e., in the other two conditions, which are called baseline conditions.

We propose that an investigation of intentional binding can provide valuable insight into a pre-reflective experience of agency particularly in OCD, for several reasons. Firstly, intentional binding is an implicit phenomenon. While its experimental operationalization requires explicit time judgements, the effect of interest—a shift in perceived time from baseline to operant conditions—remains unnoticed by participants. Compared to explicit agency judgements, intentional binding is thus assumed to be a pre-reflective phenomenon, and therefore yields insight that is complementary to, rather than confounded with, cognitive biases that are so prevalent in OCD.

Secondly, intentional binding is generally accepted as an indicator of sensed agency [44]. Furthermore, unlike physiological phenomena such as evoked responses, it directly characterizes one aspect of the phenomenology of agency itself, i.e., the subjective experience of time around an action and its consequence.

Thirdly, intentional binding is a compound phenomenon whose component processes are highly relevant to the sensorimotor theories of compulsion reviewed above. Specifically, there are predictive as well as retrospective contributions to the action binding component of intentional binding [50,51]. Moore and Haggard [50], for example, found that the perceived time of a keypress was shifted forward when participants expected a tone presented with high probability, compared to a condition in which the tone was presented with relatively low probability, and therefore less expected. This forward shift of the perceived time of action occurred irrespective of whether a tone was eventually presented or not, providing evidence for a predictive, possibly prospective, component of action binding (Figure 1B, “prospective”). However, presence vs. absence of a tone did change the perceived timing of the preceding keypress in the condition in which tones were presented with relatively low probability. Specifically, presence, compared to absence, of the tone shifted the perceived timing of the preceding keypress forward. Because the tone was presented, or omitted, only after the keypress had already been executed, this forward shift can be interpreted as evidence for a retrospective component of action binding (Figure 1B, “retrospective”). Altered function of predictive internal models in OCD, as proposed by Gentsch et al. [40], may be expected to impact on the predictive component of binding, e.g., manifest as reduced predictive (prospective) binding. An over-reliance on observable “proxies” for “internal states”, as in the SPIS model, on the other hand, may be expected to enhance retrospective action binding (see next section).

Additionally, intentional binding depends on the assignment of precision, a process which, as noted above, has received considerable attention in OCD research [9,36]. Specifically, intentional binding has been explained as a process of cue integration, where the contribution of each “cue”—the action, and its sensory consequence—to the perceived time of each event depends on its reliability, i.e., precision [52,53,54]. This, in particular, makes an investigation into intentional binding in OCD highly relevant for discriminating between alternative sensorimotor accounts of OCD, as described in the following section.

## 4. Intentional Binding as a Touchstone for Sensorimotor Theories of Compulsion

The idea that intentional binding results from cue integration [52,53,54] predicts that the binding effect should change as the precision changes with which an action and its consequence is perceived (Figure 2). Integrating information across the two “cues” seems beneficial when the perception of each cue on its own is somewhat uncertain. Put differently, a common cause of an action and subsequent sensory input (i.e., agency) is likely when the (temporal) disparity between them is not obvious, i.e., when they are perceived with some uncertainty and potential overlap [55]. Similarly, assuming as a “coupling prior” a belief of agency, that coupling prior causes stronger attraction of the two cues when their internal representations are at least somewhat mutable, i.e., uncertain, rather than highly precise [56,57].

Integration would yield little benefit, on the other hand, when perceptual representations of an action and its consequence are highly precise on their own. High precision would emphasize an existing disparity, rather than promote inference of a common cause [55], and weaken an attraction effect of any coupling prior of agency [56,57].

A key regulator of (perceptual) precision is attention (e.g., Vossel et al. [58]). De Haan et al. [25] suggest that patients with OCD pay too much attention to their own action at the time at which that action unfolds. Similarly, Belayachi and Van der Linden [26] assume a misguided focus of attention during action, specifically a focus on its motoric and mechanistic details. Finally, in line with an altered focus of attention, Levy [9] emphasizes a role of altered assignment of precision in OCD. Excessive conscious monitoring of an action, in particular its motoric or mechanistic details, as proposed by De Haan et al. [25], Belayachi and Van der Linden [26], and Levy [9], therefore implies that perception of these details becomes overly precise. Because of enhanced precision, integration of action and sensory consequence should be reduced, resulting in diminished intentional binding (Figure 2, middle row). Furthermore, this increase in precision would be directly detectable as a reduction in variance of time judgements in the intentional binding paradigm (Figure 2, middle row).

The SPIS model, on the other hand, predicts a different pattern of intentional binding. The SPIS model assumes diminished conscious access to internal states, including states of the body as sensed through proprioception, e.g., during movement [28]. Following the idea of a lost “experience of conviction” and, instead, an increase in doubt, as proposed by Shapiro [27], the SPIS effectively assumes an increase in uncertainty, or a loss of precision, as a reason for this diminished access to internal states (Figure 2, bottom row, left distributions). The SPIS model also assumes that patients compensate by enhanced attention to observable proxies, including environmental stimuli. The precision of internal representations of the tone should therefore increase (Figure 2, bottom row, right distributions). Taken together, the SPIS model predicts stronger action binding and weaker tone binding in OCD (Figure 2, bottom row), i.e., interestingly, opposite changes to action binding vs. tone binding. Importantly, the SPIS model also predicts the opposite to the excessive monitoring accounts that do not assume diminished conscious access to “internal states” (see above), specifically regarding the amount of action binding, and the precision of action time judgements. Enhanced action binding due to over-reliance on the tone, following the SPIS model, may even explain, to a certain degree, why patients with OCD assume inflated responsibility, or at least maintain that assumption. Because they serve as “proxies”, environmental stimuli may “capture” and bind the subjective experience of an action, even when that action is not the cause of that stimulus.

The SPIS also makes a clear prediction regarding retrospective binding [13]. Because of an over-reliance on the tone as an environmental stimulus serving as a proxy, the SPIS predicts strong retrospective binding (Figure 3B). Prospective (predictive) binding, on the other hand, should be diminished, given a reduced conscious access to predicted future states (Figure 3A). In contrast, excessive monitoring without diminished access, and without over-reliance on proxies, predicts a reduction in both prospective (predictive) and retrospective binding, given that both require uncertainty in perceiving an action, which is reduced due to excessive monitoring (see above; Figure 3A,B). Finally, Gentsch et al.’s [40] and Szalai’s [6] ideas of a forward model dysfunction in OCD should also reduce prospective (predictive) binding.

The intentional binding paradigm has been tested in a non-clinical cohort of university students [13], which was divided into subgroups with relatively high vs. low obsessive–compulsive tendencies, as assessed using the Obsessive–Compulsive Inventory-Revised [59]. The authors reported reduced tone binding in individuals with high vs. low obsessive–compulsive tendencies. However, this study by Oren et al. [13] has several important limitations. Firstly, the authors did not examine action binding. This is important, given that action binding, but not tone binding, discriminates the SPIS model from the idea of excessive conscious monitoring without diminished conscious access to internal states, and without over-reliance on proxies (Figure 2, bottom vs. middle row). Secondly, in both subgroups of participants, the authors observed dramatically stronger tone binding than previous studies. Specifically, tone binding in their study was around 10–12 times stronger than in the original report of intentional binding by Haggard et al. [7]. The reason for this large difference remains unclear, and raises concerns regarding plausibility of the observed effect. Thirdly, Oren et al. [13] did not dissociate prospective (predictive) and retrospective binding, despite the potential of these distinct components to disentangle sensorimotor theories of compulsion (Figure 3). They also did not examine the variability of time judgement errors as a measure of precision, even though this parameter is expected to be critically altered in OCD, compared to healthy individuals (Figure 2). Finally, the study provides no direct implications for clinical populations of OCD patients because it was restricted to a non-clinical cohort. For these reasons, Oren et al.’s [13] study does not exploit the full potential of intentional binding as an experimental paradigm that can characterize, in detail, how subjective experience of sensorimotor function is altered in OCD.

## 5. Precision, Cue Integration, Hyper-Reflexivity: Common Principles of Pathology of a Sense of Agency?

Intentional binding is altered in several neurological and psychiatric disorders. Corticobasal degeneration, for example, which may manifest with symptoms of altered experiences of action such as alien limb phenomena, is characterized by enhanced action binding [60]. Interestingly, this enhancement coincides with increased variability of the time judgement of action, in support of the idea that a core deficit in corticobasal degeneration is an imprecision in neural representations of one’s own actions, shifting a balance between time estimates for the action and the tone during cue integration and, thus, altering binding. Along similar lines, an enhancement of intentional binding in schizophrenia [61], and a shift from prospective to retrospective contributions to intentional binding [62], have also been interpretated in a context of altered precision of neural representations of action and sensory consequences [44]. Finally, functional movement disorders, previously considered a disorder of altered precision in a predictive coding/active inference framework [63], are associated with reduced intentional binding, in particular tone binding [64]. Interestingly, a “hyper-reflexivity trap” [31], as assumed in de Haan et al.’s account of compulsive action [10], has also been proposed to underlie functional movement disorder [65]. Our focus on precision in OCD, and its effects on intentional binding via cue integration (Figure 2), is therefore embedded in a broader context of clinical pathology that emphasizes precision as a potential fundamental principle of pathology of a sense of agency. Intentional binding may provide key insight not only into phenomenology, but also into principles of pathophysiology.

On the other hand, previous work in clinical disorders other than OCD points to potential confounds when studying intentional binding in OCD. Enhanced intentional binding in Parkinson’s Disease under dopaminergic medication [66] points to an involvement of the dopaminergic system in intentional binding, i.e., a neurotransmitter system also involved in the pathophysiology and treatment of OCD [67]. More than 50% of patients with OCD have psychiatric comorbidities, including mood or anxiety disorders, or tics [68,69]. While not yet formally examined, it is possible that intentional binding is altered in mood disorders, such as major depressive disorder (see, for example, a correlation with scores in the Beck Depression Inventory in Kranick et al. [64]). Zapparoli et al. [70] have shown reduced intentional binding in Gilles de la Tourette syndrome. Future experimental studies of the hypotheses developed in this perspective should therefore carefully control for psychiatric comorbidities, as well as medication, in OCD.

## 6. Limitations

We think that, for the reasons stated in the previous sections, intentional binding can provide very valuable insight into the pre-reflective subjective experience of sensorimotor function in OCD, and disentangle sensorimotor theories of OCD, with potential implications for future therapy. However, few potential limitations of our reasoning should be kept in mind. Firstly, diminished precision is but one possibility by which conscious access to internal states, such as proprioception or efferent information about imminent movement, could be compromised in the SPIS model. Indeed, Oren et al. [13] imply that a coupling prior of agency, rather than internal representations of an action, may have reduced precision in OCD, and thereby explain the diminished tone binding observed in that study. More generally, the idea that intentional binding results from cue integration has received only partial empirical support. Specifically, Wolpe et al. [53] demonstrated that action time judgements, but not tone time judgements, had higher precision in operant compared to baseline conditions. An increase in precision, however, is a hallmark of cue integration [71]. Instead, tone time judgements were less, rather than more, precise in operant conditions. Yamamoto has argued that both action and tone binding comply with mechanisms of cue integration [52]. To what extent action binding alone, or both action and tone binding, comply with rules of cue integration, remains an open question. Importantly, however, the SPIS model, and the idea of excessive monitoring without diminished conscious access, and without over-reliance on proxies, critically differ in their predictions regarding action binding, while predictions regarding tone binding are identical (Figure 2, middle vs. bottom row). The question whether tone binding, too, results from cue integration is therefore not critical for international binding to be a useful paradigm to understand mechanisms of an altered subjective experience of action in OCD.

How would neuropsychological deficits known to be associated with OCD influence performance in the intentional binding paradigm? Previous work has shown deficits in several neuropsychological domains in OCD, albeit often of small to moderate effect sizes, and of unclear clinical significances [72]. Given the task subjects face during the intentional binding paradigm, deficits in sustained attention, (spatial) working memory, and cognitive flexibility/set-shifting in OCD are particularly relevant to consider. Generally, any effects of these deficits on performance in a typical intentional binding task are expected to influence results in a way that differs substantially from the pattern predicted by the two major models discussed above, i.e., by models assuming excessive conscious monitoring of action, and by the SPIS model. Specifically, it is difficult to explain an *increase* in precision, i.e., enhanced performance, as predicted by both models, at least regarding time judgement errors for the tone (Figure 2), by a deficit in attention, or in working memory, because these deficits typically *decrease* precision, i.e., deteriorate performance. Similarly, enhanced perseverance across consecutive conditions, due to impaired set-shifting, is expected to decrease, rather than increase, precision. In addition, in typical intentional binding paradigms, the order of conditions is randomized across participants (e.g., Haggard et al. [7]), such that effects of condition order, including effects of perseverance across conditions, cannot produce systematically biasing effects at the group level. Thus, while a thorough neuropsychological assessment is necessary when testing intentional binding in OCD, typical deficits cannot explain the specific patterns of results predicted by the two models discussed above.

Finally, Moore and Haggard’s [50] operationalization of prospective binding remains potentially confounded. The effect labelled as “prospective binding” relies on differences between conditions in predictability of a tone, established via context (Figure 1A). Specifically, keypresses are followed by a tone more frequently in one condition than in the other, so that expectation of a tone is higher. However, expectation may influence action binding retrospectively, i.e., after the action has occurred. For example, when an expected tone is unexpectedly omitted, there is an omission response at the time at which that tone was expected (e.g., Sanmiguel et al. [73]), which could drive the “prospective” component of intentional binding.

## 7. Conclusions

Given the prevalence of cognitive biases in OCD, which likely influence explicit reports in response to overt inquiry, including explicit agency judgements, intentional binding can provide a useful implicit operationalization of a pre-reflective experience of agency. In light of an emerging focus on the subjective experience of sensorimotor function in OCD [6,8,9,10,24,29], this implicit operationalization can provide a touchstone for sensorimotor theories of compulsion. Specifically, we have shown how intentional binding can pit ideas of excessive monitoring of action, diminished conscious access to internal states, and over-reliance on observable proxies against each other, ideas that have distinct implications for potential future therapeutic approaches. An empirical test of the hypotheses laid out in this perspective could thus incite a novel view on a puzzling, disabling disorder.

## Figures and Tables

**Figure 1 brainsci-11-00182-f001:**
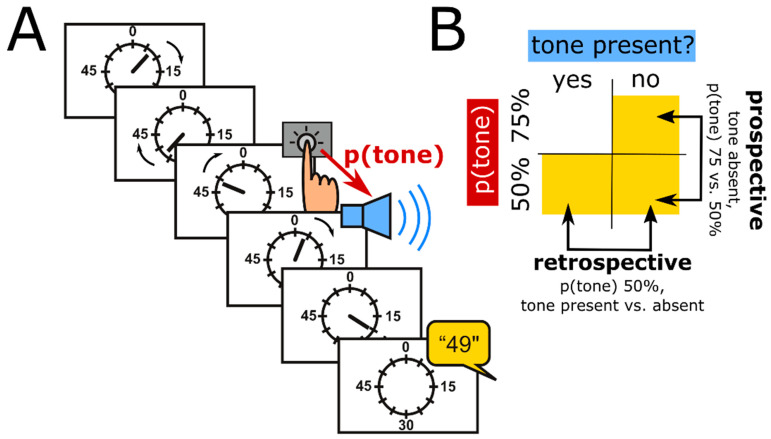
Schematic of the classic intentional binding paradigm (**A**) and design of a variant of that paradigm, which dissociates prospective and retrospective components of intentional binding (**B**). p(tone), probability of tone presentation after a keypress.

**Figure 2 brainsci-11-00182-f002:**
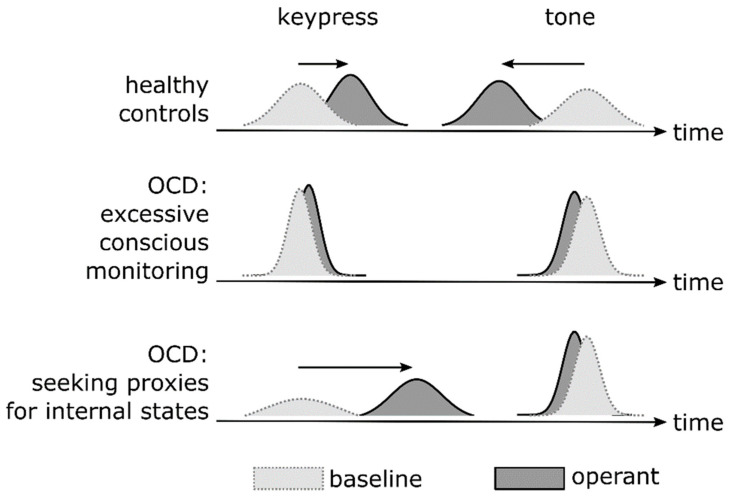
Intentional binding in healthy individuals (**top**) and patients with OCD, under the assumption of excessive conscious monitoring of action (**middle row**) and the “seeking proxies for internal states” (SPIS) model (**bottom row**). Arrows indicate changes in perceived timing from baseline (light grey) to operant conditions (dark grey).

**Figure 3 brainsci-11-00182-f003:**
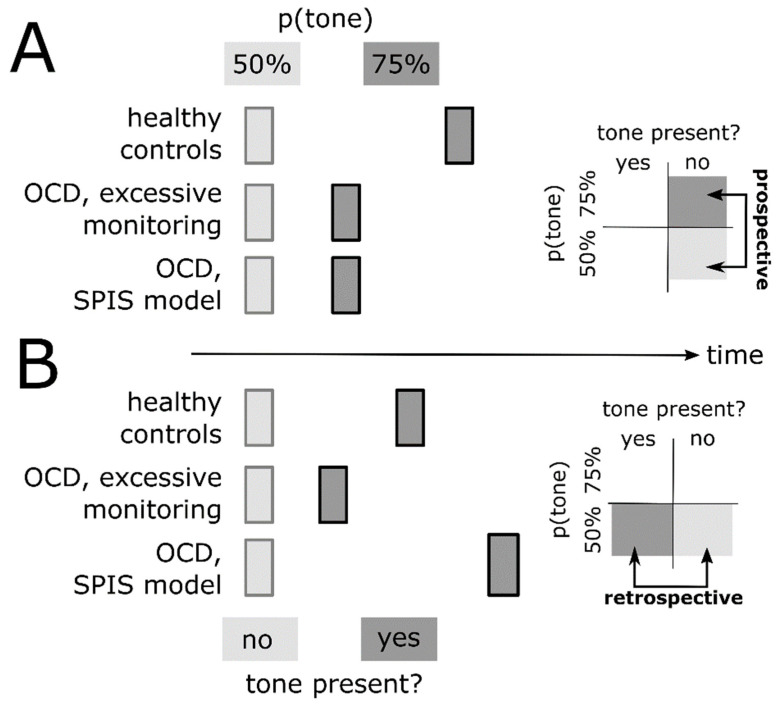
Prospective (**A**) and retrospective (**B**) components of intentional binding of action in healthy individuals (**top row**) and patients with OCD, under the assumption of excessive conscious monitoring of action (**middle row**) and the SPIS model (**bottom row**). Insets on the right represent contrasts used to isolate prospective and retrospective binding, respectively. In (**A**), the perceived time of action in the 50% tone probability condition (light grey) is aligned in time (x-axis) across groups (healthy controls vs. OCD) to illustrate group differences in a predictive (prospective) shift of the perceived time of action in the 75% tone probability condition (dark grey). In (**B**), the perceived time of action in the 50% tone probability condition without actual tone presentation (light grey) is aligned in time across groups to illustrate group differences in a retrospective shift of the perceived time of action in the 50% tone probability condition with actual tone presentation (dark grey).

**Table 1 brainsci-11-00182-t001:** Summary of theories of altered subjective experience of sensorimotor function in OCD, together with their implications for treatment.

	Altered Monitoring of Action in OCD	Implications for Treatment
Pathological Attempts to Monitor Details of Action to which Humans Normally Have Little Conscious Access	“hyper-reflexivity trap” (de Haan et al. [10,25]) low-level action identification (Belayachi & van der Linden [24,26]) altered assignment of precision (Levy [9])	attenuate conscious monitoring of action [10]
Pathologically Diminished Conscious Access to Details of Action	“Seeking proxies for internal states” (Liberman & Dar [8,12,27,28,29])	train conscious monitoring of action [12,30]

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
