# Peer review of "A Psychophysical Window onto the Subjective Experience of Compulsion"

_brainsci, 2021, doi:10.3390/brainsci11020182_

Round 1
Reviewer 1 Report
This is a very well written scholarly exploration of the euristic value of applying the intentional binding experimental paradigm to current sensorimotor theories of compusions in OCD. It is therefore bizarre that the Oren, Eitam and Dar's paper who were the first not only to propose this but also to test it experimentally is cited as reference n° 56, just before limitations.
Authors are strongly advised to do the decent thing, that is to credit the mentioned colleagues for their merit and inspiration since from the very beginning of their manuscript, that is, in the Intro and not at the end.
Needless to say I have no conflict of interest whatsoever with the above mentioned researchers
Reviewer 2 Report
The manuscript by Schmidt et al., is a relatively comprehensive research review on the potential effects of sensorimotor dysfunction in intentional binding. The authors focus on the compulsion and utilize testing-based design for potential mechanisms. There are some minor concerns:
- Consider to add a table for summary of each theory (such as in Page 4, section 2).
- In section 3, describe how cognition affect the test results, including the intentional binding with patients with cognitive decline.
- List in a new table for all studies on intentional binding and the related diseases if possible.
- Missing introduction for sensorimotor theories of compulsion.
